

# Autophagy downregulation contributes to insulin resistance mediated injury in insulin receptor knockout podocytes *in vitro*

Ying Xu[1,*], Qi Zhou[2,*], Wei Xin[3], Zhaoping Li[4], Liyong Chen[5] and Qiang Wan[6]

[1] Department of Nephrology, Shandong Provincial Hospital Affiliated to Shandong University, Jinan, Shandong, China
[2] School of Medicine, Shandong University, Jinan, Shandong, China
[3] Central Lab, Shandong Provincial Hospital Affiliated to Shandong University, Jinan, Shandong, China
[4] School of Public Health, Shandong University, Jinan, Shandong, China
[5] Department of Nutrition, Shandong Provincial Hospital Affiliated to Shandong Hospital, Jinan, Shandong, China
[6] Department of Nephrology, Qianfoshan Hospital Affiliated to Shandong University, Jinan, Shandong, China
[*] These authors contributed equally to this work.

## ABSTRACT

It is unknown whether autophagy activity is altered in insulin resistant podocytes and whether autophagy could be a therapeutic target for diabetic nephropathy (DN). Here we used shRNA transfection to knockdown the insulin receptor (IR) gene in cultured human immortalized podocytes as an *in vitro* insulin resistant model. Autophagy related proteins LC3, Beclin, and p62 as well as nephrin, a podocyte injury marker, were assessed using western blot and immunofluorescence staining. Our results show that autophagy is suppressed when podocytes lose insulin sensitivity and that treatment of rapamycin, an mTOR specific inhibitor, could attenuate insulin resistance induced podocytes injury via autophagy activation. The present study deepens our understanding of the role of autophagy in the pathogenesis of DN.

## INTRODUCTION

Diabetic nephropathy (DN) is the leading cause of end-stage kidney disease (ESRD) worldwide. It is also a major devastating complication of diabetes mellitus (DM), with up to 40% of diabetic patients experiencing this problem (*Shi & Hu, 2014*). With the rapidly increasing prevalence of DM being a major global health issue, it becomes more and more important to find therapeutic interventions directed at preventing the development and progression of DN.

The natural history of DN is dominated by progressive albuminuria, and podocytes are key components of the ultrafiltration system in the glomeruli. Moreover, podocyte number and morphology have been proved to be predictors of DN progression. This makes the glomerular podocyte an attractive early target cell. Molecular mechanisms involved in the etiology and progression of DM and its complications have been studied intensively. Among them, insulin resistance was proved to be a critical one. In glomeruli of obese and

Corresponding author
Qiang Wan,
wanyanshaoqiang@163.com

diabetic rats, insulin sensitivity was reduced (*Mima et al., 2011*). Previous studies have shown that podocytes are insulin responsive cells in glomeruli (*Madhusudhan et al., 2015*; *Tejada et al., 2008*), and a loss of podocyte insulin sensitivity in perfused glomerulus results in an albuminuric phenotype even under normal glycemic conditions (*Coward et al., 2005*). Thus, insulin signaling in podocytes is essential for normal glomerular function. This has been proved by studies that show specific deletion of podocyte insulin receptor (IR) causes significant proteinuria and glomerulosclerosis in mice (*Welsh et al., 2010*). However, how podocyte inuslin resistance leads to podocytes injury remains unclear.

Autophagy is an intracellular catabolic process by which aggregates and malfunctioned organelles are degraded to maintain intracellular homeostasis (*Levine & Ranganathan, 2010*). Defects in autophagy have been closely associated with many human diseases, including cancer, neurodegenerative disorders. Accumulating evidence suggests that regulation of autophagy system may become a new therapeutic option for treatment of DN (*Kume et al., 2014*). In both genetic and dietary mouse models of obesity and insulin resistance, decreased autophagy in hepatic cells has been observed (*Yang et al., 2010*). Moreover, in obese animals, the activation of autophagy could be protective metabolic abnormalities (*He et al., 2012*). However, it is worthy of notice that modulation of autophagy in DM varies in different cell types. In pancreatic $\beta$-cells, autophagy is activated due to peripheral insulin resistance, whereas autophagy is inhibited in the hepatic cells of Type 2 diabetes (T2D) mice with insulin resistance (*Rovira-Llopis et al., 2015*). Whether the process of autophagy in insulin resistant podocytes is altered is still an open question. There are also uncertainties about whether autophagy participates in the insulin resistance mediated podocyte injury. The present study aims at evaluating the role of autophagy in diabetic nephropathy with focus on podocyte insulin resistance. We hypothesize that blunted autophagy activity in insulin resistant podocyte is one of the mechanisms that accounts for the podocytes injury in the pathogenesis of diabetic nephropathy. To test our hypothesis, IR knockout podocytes were used as an *in vitro* insulin resistant model.

## MATERIALS AND METHODS

### Cell culture

Conditionally immortalized human podocytes were obtained from professor Fan Yi of Shan Dong University. The cells were maintained in RPMI 1640 medium (HyClone, South Logan, UT, USA) containing 10% heat-inactivated fetal calf serum (Hyclone, USA) and 100 U/ml penicillin (Hyclone, USA) in the presence of 5% $CO_2$ as described (*Saleem et al., 2002*). The cells were cultured in 33 °C to sustain podocyte proliferation, then podocytes were cultured at 37 °C for 10–14 days to induce differentiation. All experiments were performed on passages 10–14 differentiated podocytes in the present study. Insulin (200 nM) was added 30 min before cells were harvested.

### Knockdown of insulin receptor (IR) by shRNA transfection

Selected shRNA lentivirus vector (pGMLV-SC1) against IR or negative control shRNA with eGFP were designed and purchased from Genomeditech Company (Shanghai, China).

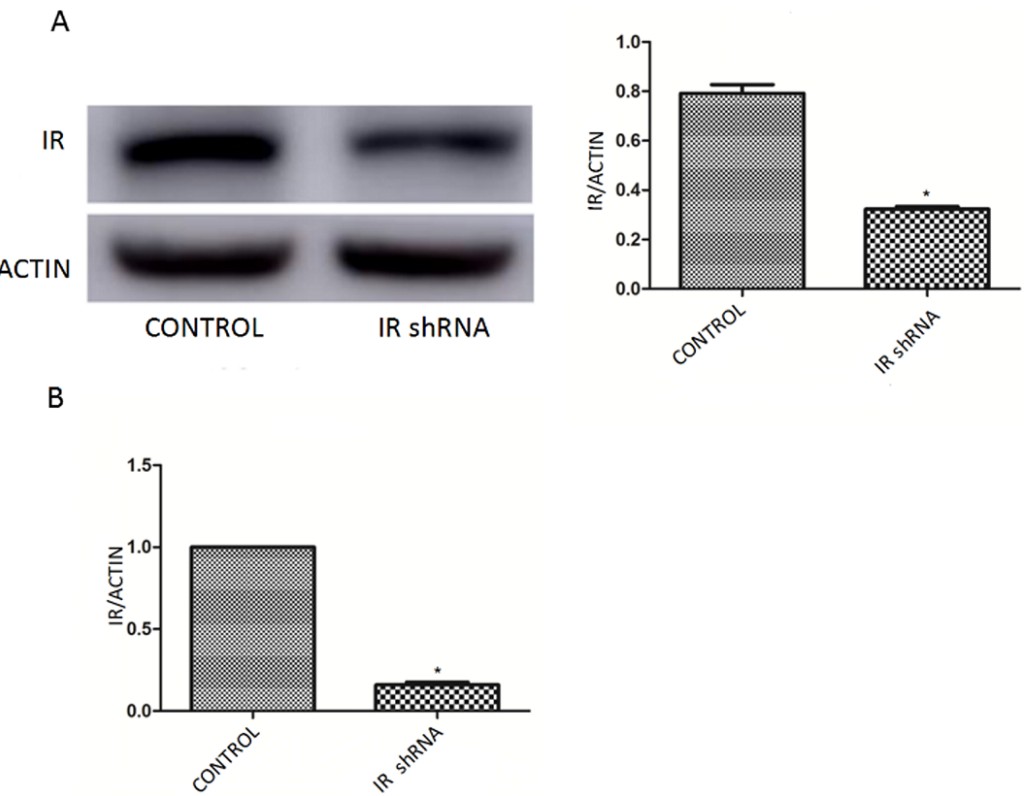

**Figure 1 Successful downregulation of IR by IR shRNA transfection.** (A) Western blots of IR after transfected with IR shRNA. (B) qRT-PCR analysis showed a 80% knockdown efficiency of IR after transfected with IR shRNA.

The transfection was performed as previously described (*Xu et al., 2014*). Figure 1 shows the successful downregulation of IR by IR shRNA transfection. Under the experimental conditions, we routinely obtain 80% downregulation of IR (Fig. 1). Thus, IR shRNA transfection was used as the *in vitro* model to study the role of autophagy in insulin resistant podocytes.

## Real time quantitative PCR (RT-qPCR)

Total RNA was isolated from podocyte and reverse transcribed to cDNA by PrimeScript$^{TM}$ RT Reagent Kit with gDNA Eraser (Takara). Real time PCR was performed using the light cycler 480 (Roche Diagnostics) with SYBR Premix Ex Taq$^{TM}$ II (Takara). The knockdown efficiency of the IR was confirmed by real time quantitative PCR (RT-qPCR) using sequence-specific primers for IR (forward 5′- GGAGCTGTCCTAGGTGCTGTTTC -3′ and reverse 5′- CTTGTGTCAGTTCCCACAGCTTC -3′), which were designed by TaKaRa. The expression levels of IR were normalized to $\beta$-actin expression level (forward 5′- TGGC ACCCAGCACAATGAA -3′ and reverse 5′- CTAAGTCATAGTCCGCCTAGAAGCA -3′).

## Western blot analysis

The procedure of western blot analysis was carried out as described previously (*Delfin et al., 2011*). The following primary antibodies were used: antibodies for SQSTM1/p62 (5114; Cell

Signaling), LC3B (ab48394; Abcam), Beclin-1 (SC-11427; Santa Cruz), IR (sc-711; Santa Cruz) and nephrin (ab58968; Abcam) Primary antibody against $\beta$-actin and horseradish peroxidase-conjugated secondary antibodies were from ZSGB-BIO.

### Immunofluorescence staining

The immunofluorescence staining was performed using protocol modified from previous publication (*Ning et al., 2011*). Briefly, cells were fixed in 4% paraformaldehyde for 10 min at room temperature, then washed and permeabilized using PBS containing 0.3% Triton X-100, followed by blocking with 5% BSA in PBS. 1:100 dilution primary antibodies were added to the cells. After incubating overnight at 4 °C, cells were washed. 10 μg/ml Alexa Fluor488 or Alexa Fluor594 secondary antibodies were added and incubated for 30 min. Images were acquired by immunofluorescence microscopy (Nikon Ti-S, Tokyo, Japan).

### Electron microscopy

After treatments, podocytes were trypsinized and collected into centrifuge tubes after washing by PBS. Then, the cells were fixed by 3% glutaraldehyde at 4 °C, dehydrated by dimethylketone. After embedment in Epon-812, the samples were cut into ultrathin sections (70 nm). Uranium acetate and plumbum citrate were used to dye the ultrathin sections. The samples were observed with JEM-100sX electron microscopy.

### Statistics

Data are presented as means ± SD unless stated otherwise. Data were analyzed using repeated measurement ANOVA followed by $t$ test when appropriate with two tailed $p$-values < 0.05 considered statistically significant. GraphPad Prism software (version 5) was used for data analysis.

## RESULTS

### Autophagy activity was down-regulated in IR-knockdown podocytes

To investigate autophagy activity under insulin resistant conditions, we examined the changes of autophagy related protein abundances in IR deficient podocytes (Fig. 1). Figure 2 shows the western blots of Beclin1, p62 and LC3II, indicating that autophagy was downregulated in IR knockdown podocytes. During the autophagosome formation, LC3-I is processed into a lapidated LC3-II form. The LC3-II/LC3-I ratio is considered as a marker of auophagosome formation. Chloroquine (CQ) was used as the lysosomal activity blocker to evaluate the autophagic flux. Compared with control group, a significant less LC3-II was detected in the IR-knockdown cells (Fig. 2A). Similar to the LC3 protein, western blots showed a decreased level of Beclin1 (a marker for autophagosome initiation complex) (Fig. 2B). Compared to control cells, IR knockdown podocytes showed significant increased expression of p62, indicating decreased autophagy activity (Fig. 2B). Figure 3 shows the immunofluorescence of Beclin1, p62 and LC3. Consistent with the western blot results, the immunofluorescence staining of Beclin1 and p62 also indicates a decreased autophagy activity in IR-knockdown podocytes. Figure 4 shows the electron microscopy images of control and IR knockdown podocytes. There were significantly less autophagosomes in IR knockdown podocytes compared to control cells.

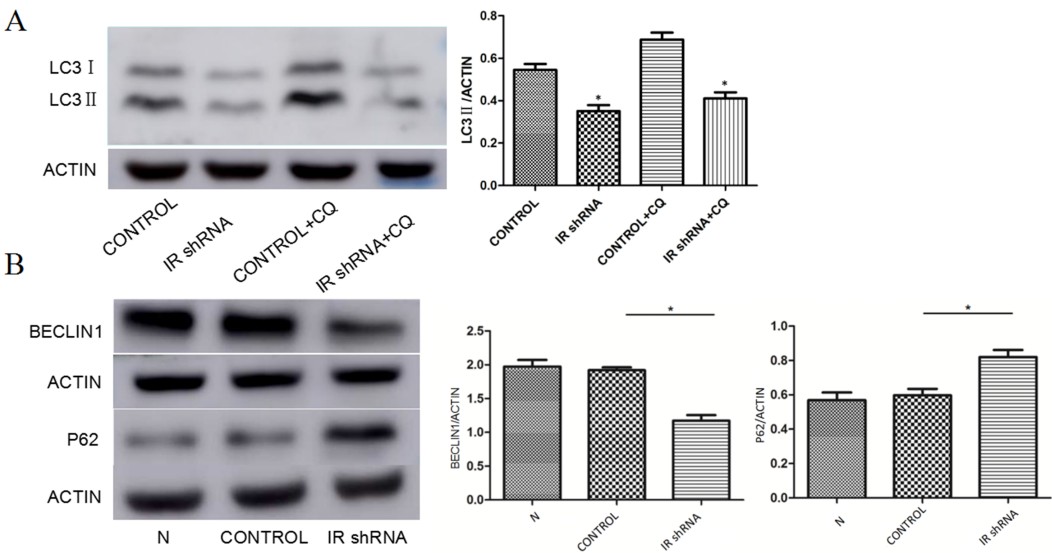

**Figure 2 Autophagy was down-regulated after knockdown of IR.** (A) After 200 nM insulin stimulation for 30 min, the expression of LC3ll were decreased in cells transfected with IR shRNA with or without 50 uM chloroquine (CQ), compared with control shRNA. *$P < 0.05$ vs. control shRNA. (B) After 200 nM insulin stimulation for 30 min, the expression of BECLIN1 were decreased, but P62 was up-regulated in cells transfected with IR shRNA, compared with control shRNA. *$P < 0.05$ vs. control shRNA.

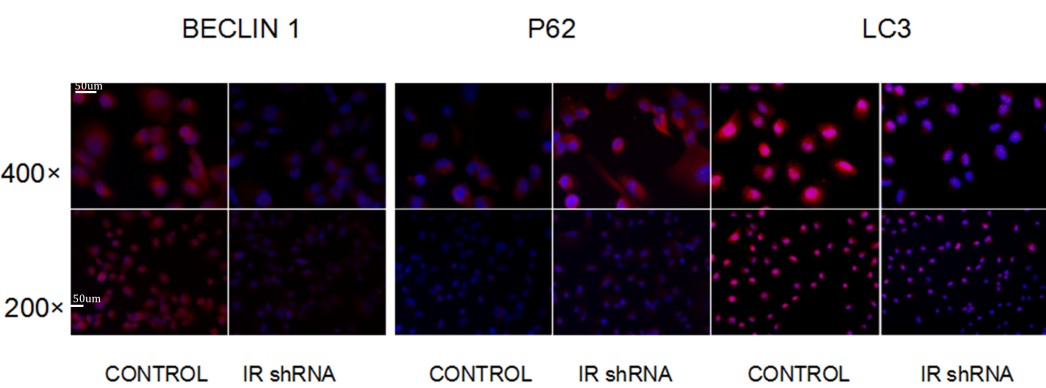

**Figure 3 Immunofluorescence of autophagy markers shows that autophagy was down-regulated after knockdown of IR.** After 200 nM insulin stimulation for 30 min, the staining of BECLIN1 and LC3ll was decreased but P62 was enhanced in cells transfected with IR shRNA, compared with cells transfected with control shRNA.

## IR deficiency induced podocyte injury *in vitro*

Nephrin has been proved to be critical for the action of insulin on podocytes (*Coward et al., 2007*) and its expression has been used as the marker of podocyte integrity. Figure 5 shows the nephrin expression levels in control and IR knockdown podocytes. Both western blot (Fig. 5A) and immunofluorescence staining (Fig. 5B) of nephrin show that IR-knockdown podocytes had a significant decrease of nephrin expression level, indicating podocyte injury induced by insulin resistance.

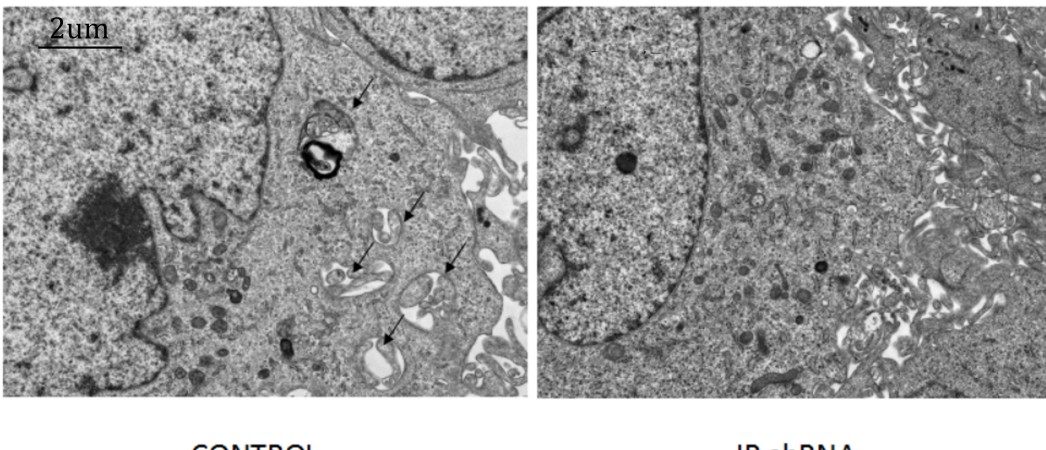

**Figure 4 Electronic microscopy analysis of podocyte shows that autophagy was down-regulated after knockdown of IR.** After 200 nM insulin stimulation for 30 min, the number of autophagosomes in cells transfected with IR shRNA was decreased compared with cells transfected with control shRNA. Black arrows indicate autophagosomes. Magnification ×15,000.

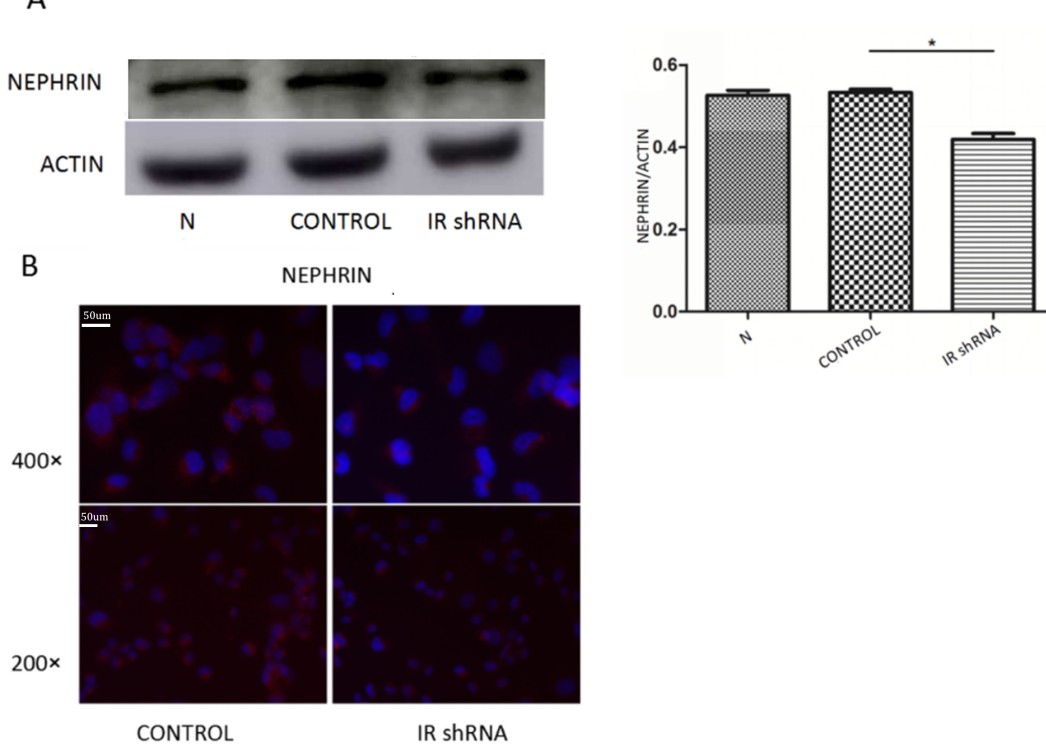

**Figure 5 Nephrin expression was down-regulated after knockdown of IR.** (A) WB showed after 200 nM insulin stimulation for 30 min, the expression of NEPHRIN was down-regulated in cells transfected with IR shRNA, compared with cells transfected with control shRNA *$P < 0.05$ vs. control shRNA. (B) Immunofluorescence demonstrated that after 200 nM insulin stimulation for 30 min, NEPHRIN staining was decreased in cells transfected with IR shRNA, compared with cells transfected with control shRNA.

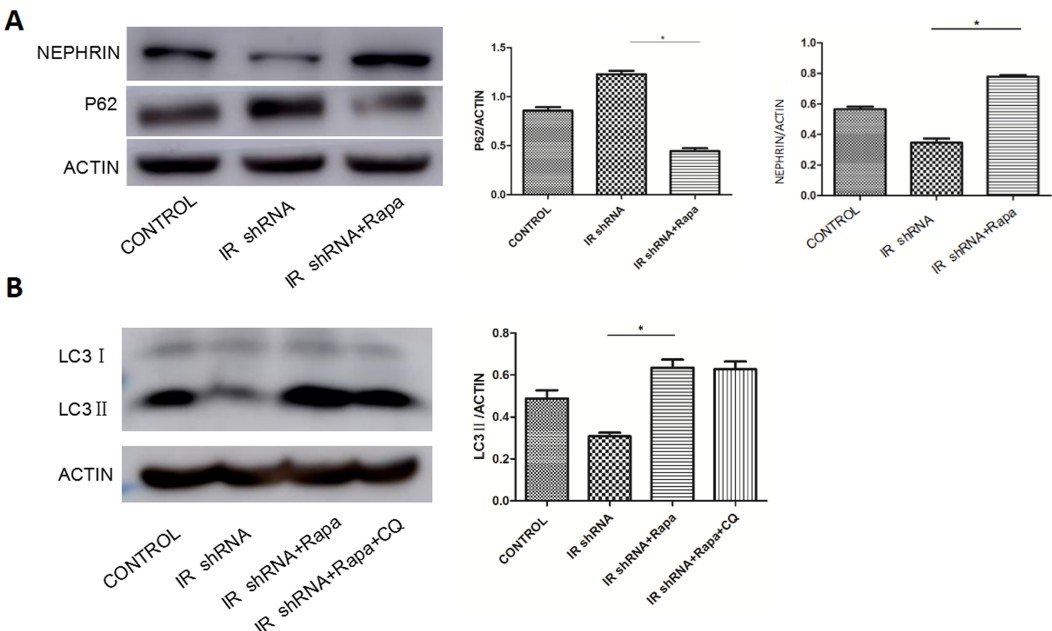

**Figure 6** **Rapamycin (RAPA) activated autophagy in IR-knockdown podocytes and increased the expression of Nephrin.** (A) After 10 uM Rapamycin (RAPA) stimulation for 2 h, the expression of p62 in IR-knockdown podocytes were decreased while NEPHRIN was up-regulated compared with cells without RAPA. *$P < 0.05$ vs. IR shRNA.(B) After 10 uM Rapamycin (RAPA) stimulation for 2 h the expression of LC3ll in IR-knockdown podocytes was up-regulated in cells with or without CQ compared with cells without RAPA. *$P < 0.05$ vs. IR shRNA.

## Rapamycin ameliorated insulin resistance induced podocyte injury via autophagy activation

Rapamycin has been clinically used to inhibit rejection after organ transplantation as a FDA approved immunosuppressive drug. It works by binding to mammalian target of rapamycin (mTOR) and inhibiting the activity of mTOR protein kinase (*Hartford & Ratain, 2007*). Rapamycin has been shown to induce autophagy in many cell types and species (*Huber, Walz & Kuehn, 2011*). Figure 6 shows the p62, LC3II and nephrin level in IR-deficient podocytes with or without the treatment of rapamycin. DMSO was used as the control group since rapamycin was resolved in DMSO. Compared to the DMSO control group, IR-deficient podocytes treated with rapamycin showed elevated autophagy activity indicated by increased LC3II expression. Meanwhile, nephrin expression was restored by rapamycin treatment. Our results indicate that podocyte injury under insulin resistant conditions was ameliorated by rapamycin treatment via activation of autophagy.

## DISCUSSION

The main findings of the current study are that under insulin resistant conditions, autophagy activity in podocytes was suppressed *in vitro*; Treatment of rapamycin, an mTOR specific inhibitor, could attenuate insulin resistance induced podocytes injury via autophagy activation.
Our results show that autophagy related proteins such as Beclin1 and LC3 were down regulated and p62 was increased in IR-knockdown podocytes. Thus, our data indicates decreased autophagy activity is induced in podocytes under insulin resistant conditions. It has been demonstrated previously that autophagy plays a renoprotective role in the kidney. Podocyte-specific autophagy deficient mice generated by *Atg5* gene deletion have glomerular lesions accompanied by podocytes apoptosis and albuminuria with aging (*Mizushima et al., 2004*). Greatly increased susceptibility to glomerular disease was exhibited in mice lacking *Atg5* gene in podocytes. These findings underscore the importance of autophagy regulation as a critical mechanism to maintain podocyte homeostasis. With the evidence of autophagy in kidney health and disease accumulating, studies to investigate the role of autophagy in DN have attracted intensive interests. Histone deacetylase 4 inhibition could ameliorates podocyte injury and attenuates glomerulopathy in DN, and the maintenance of autophagy in podocytes was suggested to be the mechanism underlying (*Wang et al., 2014*). In patients and rats with diabetes, insufficient podocyte autophagy was observed histologically. Moreover, podocyte-specific autophagy-deficient mice developed podocytes loss and massive proteinuria in a high-fat diet-induced diabetes model that usually presented with minimal proteinuria (*Tagawa et al., 2016*). These all underline the role of autophagy in podocytes in DN pathogenesis. Our study contributes to the understanding of how autophagy activity responses to insulin resistance with further detailed mechanisms to be investigated.

Podocyte insulin resistance is associated with glomerular podocyte dysfunction. In this study, downregulated autophagy activity was demonstrated to participate in podocyte injury. Since insulin resistance is a prevalent metabolic feature in DM, it is difficult to answer the question that which happens first, the onset of insulin resistance or alterations of autophagy. High-fructose feeding mice showed disruption of autophagy in the liver, which appeared as an early event preceding the onset of insulin resistance. Drugs able to restore the autophagic flux could indeed prevent insulin resistance (*Wang et al., 2015*). Importantly, a recent study found that autophagy regulates muscle glucose homeostasis and increase insulin sensitivity (*Tam & Siu, 2014*). However, most of the works about autophagy and insulin resistance were done in adipose tissue, skeletal muscles or liver, and what the relationship in the kidney remains unknown. Our results show that after IR knockdown in podocyte, there are decreased autophagy activities. With the current study, we report a decreased autophagy that was poised in an *in vitro* setting of insulin resistance. Hence, a reciprocal interaction between podocyte insulin resistance and autophagy activity should be proposed. Taken together, among mechanisms that might be altered in podocytes under insulin resistant conditions, autophagy seems to play a role. Other mechanisms such as oxidative stress and endoplasmic reticulum (ER) stress are also implicated. Recently, it has been shown that ER stress (*Madhusudhan et al., 2015*) and mitochondrial function (*Ising et al., 2015*) are modulated by insulin signaling. In adipocytes *in vitro*, endoplasmic reticulum (ER) stress causes insulin receptor (IR) down-regulation, which accounts for insulin resistance. While inhibition of autophagy could alleviate the ER stress induced IR down-regulation (*Zhou et al., 2009*). It is also proposed that in diabetes, autophagy system is activated in response to ER stress induced insulin resistance (*Zhang et al., 2015*). These

results indicate that autophagy regulation in DN is a rather complicated process. The genes and signaling pathways that link autophagy and insulin resistance as well as podocyte injury *in vivo* need to be delineated.

Our results showed that with manipulation, the activation of autophagy could protect the podocyte from injuries induced by insulin resistance. Podocyte insulin resistance has been considered to play a role in genesis and propagation of DN. The study of insulin signaling pathways in podocyte has gained considerable momentum recently. Our results show that treatment of rapamycin could attenuate insulin resistance induced podocyte injury via autophagy activation. Rapamycin is an mTOR specific inhibitor, while mTOR is an evolutionarily conserved serine/threonine kinase. The role of rapamycin in DN has been studied in a large spectrum. Several recent studies have shown that mTORC1 signaling is highly activated in podocytes of diabetic kidneys in human beings and animals. Studies by others show that rapamycin ameliorated renal hypotrophy in mice model of diabetes (*Sakaguchi et al., 2006*; *Sataranatarajan et al., 2007*). Our previous study also found that rapamycin attenuated high glucose induced lipotoxicity and epithelial-to-mesenchymal transition (EMT) via autophagy activation in proximal tubular cells (*Xu et al., 2015*). Taken together, the present study further provides evidence for use of rapamycin to treat DN.

In summary, we found under insulin resistant conditions, autophagy activity in podocytes is downregulated *in vitro*, and activation of autophagy could prevent insulin resistance induced podocyte injury. The present study deepens our understanding of the role of autophagy in the pathogenesis of DN. However, since the development of DM and/or DN is a very complex process, autophagy may have impact on each stage in the pathogenesis. Future research is demanded to further clarify the roles of autophagy, which will provide more evidences for autophagy to be a therapeutic target to prevent or alleviate development of DN.

### Funding
This work was performed in the central laboratory supported by Shandong Provincial Hospital Affiliated to Shandong University and supported by the National Natural Science Foundation of China (81200610, 81441106, 81570654 to QW, 81471007 to WX, 81500553 to YX, 81470498 to LC), Grant BS2015YY018 from Shandong Doctoral Foundation of China to YX, and Grant 201311022 from Jinan Science and Technology Developing Project to WX. The funders had no role in study design, data collection and analysis, decision to publish, or preparation of the manuscript.

### Grant Disclosures
The following grant information was disclosed by the authors:
Shandong Provincial Hospital Affiliated to Shandong University.
National Natural Science Foundation of China: 81200610, 81441106, 81570654, 81471007, 81500553, 81470498.
Shandong Doctoral Foundation of China: BS2015YY018.
Jinan Science and Technology Developing Project: 201311022.

## Competing Interests

The authors declare there are no competing interests.

## Author Contributions

- Ying Xu conceived and designed the experiments, analyzed the data, wrote the paper, reviewed drafts of the paper.
- Qi Zhou performed the experiments, analyzed the data, prepared figures and/or tables, reviewed drafts of the paper.
- Wei Xin contributed reagents/materials/analysis tools, reviewed drafts of the paper.
- Zhaoping Li and Liyong Chen contributed reagents/materials/analysis tools.
- Qiang Wan conceived and designed the experiments, reviewed drafts of the paper.

## Data Availability

The research in this article did not generate any raw data.

## Supplemental Information

Supplemental information for this article can be found online at http://dx.doi.org/10.7717/peerj.1888#supplemental-information.

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
