# Peer review of "Autophagy downregulation contributes to insulin resistance mediated injury in insulin receptor knockout podocytes in vitro"

_PeerJ, doi:10.7717/peerj.1888_

## Round 0.1 · original submission · Major Revisions

Both Referees consider your manuscript of interest. Please carefully adress all points raised by them, in particular those concerning the analysis of autophagy flux by using lysosome inhibitors.

Reviewer 1 ·

Basic reporting

Insulin resistance is a critical complication in the progression of diabetic nephropathy. Podocytes, key components of the ultrafiltration system in the glomeruli, are insulin responsive cells in glomeruli and insulin signaling in podocytes is essential for normal glomerular function. Podocytes insulin resistance is associated with glomerular podocyte dysfunction.
How podocytes insulin resistance leads to podocytes injury remains unclear. In this manuscript, the authors hypothesize that reduced autophagy activity in insulin resistance podocyte participate in podocyte injury. They reported that activation of autophagy, by rapamycin treatment, could attenuate podocyte injury caused by insulin resistance.
The article includes adequate introduction and background to evaluate the work into the specific context of interest, but some relevant prior articles on this topic should be referenced ( Tagawa et al., (2015)Impaired podocyte autophagy exacerbates proteinuria in diabetic nephropathy. Diabetes. 2015 Sep 17).
This paper contains interesting information and is potentially acceptable for publication. However, there are a number of concerns that need to be addressed by the authors.

Experimental design

Major concerns
Fig. 2 The authors analyse Beclin-1, p62 and LC3 protein levels using Western blotting analysis. Western blot analysis clearly show a decrease in Beclin1 protein levels and a significant increase expression of p62. However, according to the published guidelines for monitoring autophagy the authors should analyse autophagy activity by evaluate LC3II/actin ratio instead of LC3II/LC3I ratio and analyse autophagic flux by comparing LC3 levels in the presence or absence of lysosomal inhibitors.

Fig.3 In accordance with western blot results, the authors show immunofluorescence staining of Beclin1 and p62. To further confirm these results, the authors should perform an immunofluorescence staining for LC3.

Fig. 6 The authors analyse LC3 protein lelevs using Western blotting analysis. To verify that rapamycin lead to autophagy activation, according to the published guidelines for monitoring autophagy the authors should analyse autophagy activity by evaluate LC3II/actin ratio instead of LC3II/LC3I ratio and analyse autophagic flux by comparing LC3 levels in the presence or absence of lysosomal inhibitors.

Minor concerns
Figure legends sections are very little informative and should be improved. Figures lack important details:
Fig.6 : the quantification graphic legend lacks the rapamycin point
Fig. 2-3: the protein name Beclin should be replaced with Beclin1

Validity of the findings

Fig. 5 The authors report a decrease of nephrin expression in IR knockdown podocytes, as indication of podocyte injury induced by insulin resistance. The authors hypothesize that reduced autophagy activity in insulin resistance podocyte participate in podocyte injury, to confirm this hypothesis the authors should analyse nephrin expression levels after silencing autophagic genes, such as Beclin1 or ATGs.

Reviewer 2 ·

Basic reporting

No comments

Experimental design

No comments

Validity of the findings

No comments

Additional comments

The manuscript of Xu Ying et al. describes a reduction of autophagy in human podocytes where the insulin resistance was carried out by downregulating the expression of Insulin Receptor (IR). To analyze the occurrence of autophagy activation, authors analyzed LC3I/LC3II conversion, p62 and BECLIN1 levels by western blotting, all markers commonly used to monitor autophagy. They also performed immunofluorescence staining of p62 and BECLIN 1 and Electron Microscopy to observe autophagic vesicles. To monitor podocytes integrity they analyzed Nephrin levels. Altogether, authors demonstrate that autophagy is impaired in their system using several approaches although some important points have to be solved to assess a publication on PeerJ.
Major points:
- It was demonstrated that Endoplasmic Reticulum (ER) stress associated to insulin resistence cause a downregulation of IR by autophagy (Zhou et al., 2009). In the light of these, it will be important to consider that the direct downregulation of IR could affect the proper autophagic response associated to insulin resistence. In the light of these, authors should consider other approaches to promote insulin resistence or discuss properly their choice and considering a revision of the title.
- Introduce figure legends in the manuscript.
- Despite authors found a reduction of autophagy following IR shRNA, high levels of LC3II are reported both in figure 2 and 6. These results will be more convincing by blocking the lysosomal activity to properly evaluate the autophagic pathway.
- In figure 6 the analysis of p62 levels should be included.
Minor points:
- Some sentences, especially in the discussion session, are not very well understandable. Consider a revision.
- In figure 5A improve the quality of the Nephrin western blot.
- Figure 1 is not described in the result session.
- EMT acronym is not described.

---

## Round 0.2 · Minor Revisions

Before manuscript acceptance, it is required that you modify the manuscript following Referee 2 suggestions.

Reviewer 1 ·

Basic reporting

In the reviewed manuscript the authors significantly improved experimental results and added the necessary information in the discussion. Suggested changes to the figures and additional experiments have been done in accordance with the referees requests. Based on these considerations, the manuscript is acceptable for publication.

Experimental design

no comments

Validity of the findings

no comments

Reviewer 2 ·

Basic reporting

No comments

Experimental design

No Comments

Validity of the findings

No Comments

Additional comments

The revised manuscript from Xu Ying et al. was significantly improved from the previous version, in particular the addition of lysosome inhibitors to evaluate autophagy strengthen all presented results. All concerns from this reviewer were addressed, although few minor points have to be develop to be suitable to publish on PeerJ:
- Magnifications reported in figure 4 have to be similar and a scale bar have to be added to EM and fluorescence microscopy images.
- In new line 128 add “deficient podocytes (figure 1). Figure 2 shows the western blots of Beclin 1….”
- Figure 2A and B should be inverted in order to follow the arrangement reported in the results.
- All references have to be check, there is some missing citation reported in the text but not present in the list and some formatting problems.

---

## Round 0.3 · accepted · Accept

The referees' concerns have been addressed.